# The Role of Formyl Peptide Receptors in Permanent and Low-Grade Inflammation: *Helicobacter pylori* Infection as a Model

**DOI:** 10.3390/ijms22073706

**Published:** 2021-04-02

**Authors:** Paola Cuomo, Marina Papaianni, Rosanna Capparelli, Chiara Medaglia

**Affiliations:** 1Department of Agriculture Sciences, University of Naples Federico II, Portici, 80055 Naples, Italy; paola.cuomo@unina.it (P.C.); marina.papaianni@unina.it (M.P.); 2Department of Microbiology and Molecular Medicine, University of Geneva Medical School, 1211 Genèva, Switzerland; chiara.medaglia@unige.ch

**Keywords:** *Helicobacter pylori*, formyl peptide receptors, inflammation

## Abstract

Formyl peptide receptors (FPRs) are cell surface pattern recognition receptors (PRRs), belonging to the chemoattractant G protein-coupled receptors (GPCRs) family. They play a key role in the innate immune system, regulating both the initiation and the resolution of the inflammatory response. FPRs were originally identified as receptors with high binding affinity for bacteria or mitochondria N-formylated peptides. However, they can also bind a variety of structurally different ligands. Among FPRs, formyl peptide receptor-like 1 (FPRL1) is the most versatile, recognizing N-formyl peptides, non-formylated peptides, and synthetic molecules. In addition, according to the ligand nature, FPRL1 can mediate either pro- or anti-inflammatory responses. Hp(2-20), a *Helicobacter pylori*-derived, non-formylated peptide, is a potent FPRL1 agonist, participating in *Helicobacter pylori*-induced gastric inflammation, thus contributing to the related site or not-site specific diseases. The aim of this review is to provide insights into the role of FPRs in *H. pylori*-associated chronic inflammation, which suggests this receptor as potential target to mitigate both microbial and sterile inflammatory diseases.

## 1. Introduction

The human body is continuously exposed to millions of pathogens, which can enter it through different ways. The immune system, and in particular, innate immunity, has an important role in controlling the infections, representing the front-line defense against invading pathogens [1,2,3].

Innate immunity is mediated by anatomic barriers (skin and mucosa), physiological functions (pH values, temperature, chemical mediators), and cellular elements [4]. Skin and mucosal surfaces—including the gastrointestinal tract, the respiratory tract, the urogenital tract, and ocular surface—are the earliest sites of host–pathogen interactions, separating the body from the external environment [3,5]. This feature makes them directly involved in preventing the entry of invading microorganisms [3], acting as chemical–physical barriers [6]. Nevertheless, mechanical or chemical cleansing mechanisms provided by anatomic barriers can fail, promoting the action of other innate immune elements—specifically, immune cells, which participate to potentiate the protective role of the mucosal surface. The mucosal immune system, indeed, comprises approximately 80% of all the immune cells (macrophages, monocytes, neutrophils, dendritic cells, B and T cells, and natural killer cells) [6,7], which express specific receptors called pattern recognition receptors (PRRs).

Pattern recognition receptors are germline-encoded receptors, evolutionary selected and conserved. They mediate innate immune recognition by detecting conserved molecular structures associated with pathogens or host tissue/cell damage, also called pathogen-associated molecular patterns (PAMPs) [8] or damage-associated molecular patterns (DAMPs), respectively.

The activation of pattern recognition receptors initiates the inflammatory response, leading to cytokine, chemokine, prostaglandin, and leukotriene production, as they are key messengers signaling the occurrence of tissue damage.

Inflammation is a complex biological response initiated by the host against a microbial infection or an injury. The aim of this process consists of protecting the host by removing the dangerous stimulus and restoring the damaged tissue, thus favoring the return to the homeostasis [9]. However, if not controlled and self-limited, the inflammatory response may cause severe tissue damage, increasing the microbial pathogenicity [10,11] and contributing to the development of chronic inflammatory diseases [12].

Formyl peptide receptors (FPRs) are a family of pattern recognition receptors, able to recognize principally N-formyl methionine-containing peptides, derived both from bacteria (PAMPs) and mitochondria (DAMPs) [13]. In addition to pattern recognition receptors, formyl peptide receptors regulate the inflammatory reaction by modulating the host defense with regulatory functions in both the initiation and the resolution of inflammation. This suggests their involvement in physiological as well as pathological conditions, making them an important element of the innate immune system with a critical role in human health.

The present review focuses on formyl peptide receptors of the gastrointestinal mucosal immune system, their role in inflammation associated with *Helicobacter pylori* infection, and the possibility of targeting them to modulate *H. pylori*-associated chronic inflammation.

## 2. Formyl Peptide Receptors

### 2.1. Formyl Peptide Receptors: Cell Distribution and Classification

Formyl peptide receptors are a family of classical chemoattractant receptors. Human cells present three different isoforms of FPRs: FPR1, FPR2, or formyl peptide receptor-like 1 (FPRL1), and FPR3 or Formyl peptide receptor-like 2 (FPRL2). Each receptor is encoded by a specific gene of the human FPR gene family (FPR1, FPR2, FPR3) clustered on the chromosomal region 19q13.3 [14]. FPR1, FPR2, and FPR3 genes share high level of sequence homology [9], resulting in receptor similarity. Compared to FPR1, FPRL1 has one additional amino acid, for a total of 351 residues, and a sequence homology corresponding to 69% between the two receptors. Meanwhile, compared to FPR1, FPRL2 has two additional amino acids, for a total of 352, and a sequence homology of 56% [14]. Despite the high sequence identity among the three FPRs, they differ in cell expression, role (Table 1), and ligand recognition (Table 2). Phagocytic cells were the first in which FPRs were identified. Monocytes express the three FPR genes, but their expression can change in differentiating monocytes. Specifically, monocytes differentiating in macrophages maintain FPRL1 expression, while the transition from immature to mature dendritic cells determines the loss of FPR1. The role of FPRs in macrophage and dendritic cell differentiation has been widely investigated, demonstrating that WKYMV (an FPR ligand) negatively regulates LPS-induced dendritic cell differentiation [15,16], while Lipoxin-A4, LL-37, and serum amyloid A protein (SAA) (FPRL1 agonists) mediate macrophage polarization to the anti-inflammatory form M2 [16]. These findings suggest that FPRs play a regulatory function in immune cell differentiation and activation. Unlike monocytes, neutrophils express both FPR1 and FPRL1, but they lack the FPRL2 gene [14]. Recently, adaptative immune cells were discovered to express FPRs. FPRL1 was found in human tonsillar follicular helper T cells, Th1 cells, Th2 cells, Th17 cells, and in naïve CD4 T cells (CD3+, CD4+, CD45RA+, CD45RO-, and CCR7+) [16,17].

In addition, FPR expression has also been described in platelets, microglial cells, epithelial cells, hepatocytes, fibroblasts, and endothelial cells, as well as spleen, brain, placenta, liver, and further tissues, demonstrating the almost ubiquitous distribution of these receptors [18]. The wide expression of formyl peptide receptors in immune and non-immune cells, including various organs and tissues, suggests their involvement in multiple biological functions, with inflammation as the core.

Among the three receptors, formyl peptide receptor-like 1 is the most ubiquitous one. It has been found in different non-myeloid–endothelial cells, epithelial cells, as well as the hepatocyte-and tissue-gastrointestinal tract, brain, spleen, female organ tissues, pancreas, and endocrine glands [19,20]. Furthermore, compared to FPR1 and FPRL2, FPRL1 can interact with the highest variety of chemically different ligands, including non-formylated peptides, organic molecules, and lipid mediators [19]. These properties make FPRL1 the most promiscuous formyl peptide receptor, and more generally, G-protein coupled receptor (GPCR), able to elicit different cell responses, with inflammation as the common feature.

### 2.2. FPRs: G-Protein Coupled Receptors

FPRs belong to the family of G-protein coupled receptors (GPCRs), the largest class of cell surface seven-transmembrane proteins. Conventionally, GPCRs are located on the cell surface. However, they can also be expressed in the cell nucleus [46]. Recent studies have demonstrated the nuclear localization of FPRL1 in cancer cell lines [46], providing a further element confirming the versatility of this receptor.

More than 3% of the human genome encodes 800 GPCRs, mainly involved in regulating physiological processes by responding to endogenous ligands, such as hormones, neurotransmitters, chemokines, and calcium ions, or to exogenous ligands, such as odorant molecules and light photons [47,48]. Like hormones and neurotransmitter receptors, GPCRs regulate the communication between the inside and outside of the cells [49], transducing the extracellular signals in cellular response.

GPCRs are coupled to heterotrimeric G proteins, consisting of Gα, Gβ, and Gγ subunits, which initiate the downstream signaling associated with the receptor. Ligand binding stimulates GPCR conformational change and the activation of the heterotrimeric G proteins, resulting in guanosine triphosphate (GTP)-bound Gα and Gβ-γ subunit dissociation. In the active state, G proteins activate specific enzymes, which in turn generate second messengers, leading to cellular response [50].

Chemiotaxis are a common cellular response initiated by GPCRs, expressed on chemotactic cells. This subfamily of G protein-coupled receptors plays an important role in sensing chemoattractant molecules [51], coordinating cell migration at the damaged site, and initiation of the immune response.

N-formylated peptides are one of the first identified chemotactic stimuli [14], exhibiting high affinity for G protein-coupled receptors [51]. This has been demonstrated by the observation that the pertussis toxin inhibits formyl peptide-induced chemotaxis, altering their binding affinity for the receptor [52]. More specifically, the pertussis toxin acts by inducing ADP-ribosylation of the α subunit of a Gi-heterotrimeric G protein [53], affecting the molecular shifts responsible for the cell signal translation pathway and specifically inhibiting cell migration.

Based on this evidence, FPRs are Gi protein-coupled receptors—belonging to the γ-group of rhodopsin-like receptors—able [54] to recognize the presence of bacteria or host cell damage and promote the immune response by activating the chemiotaxis.

### 2.3. N-Formylated Peptides: Signal Peptides Detected by FPRs

The capability to distinguish between self and non-self is fundamental for the wellness of the host. The immune system has the essential role of initiating a defensive response against pathogens, by detecting microbial chemical elements: pathogen-associated molecular patterns (PAMPs). Bacterial signal peptides represent an important class of PAMPs, responsible for innate immune system activation by interacting with specific pattern recognition receptors (PRRs). They play a critical role in nascent protein translocation from the cytoplasm to other sites inside or outside the cell [55,56].

Signal peptides at the N-terminal of the newly synthesized proteins are recognized by specific particles responsible for locating the proteins at the correct site. After translocation, signal peptides are removed by specific enzymes and the mature protein is released [56,57]. Prokaryotic and eukaryotic signal peptides present three highly conserved domains, containing the correct information for protein translocation and removal: an N-terminal region, starting with a methionine, a hydrophobic core, and a C-terminal region [58]. However, all prokaryotes start the protein synthesis with an N-formyl methionine [59], resulting in the fact that bacterial signal peptides are “N-formylated peptides”. Mitochondria, as bacterial ancestry [60], also initiate proteins synthesis with a formylated methionine, revealing that, besides bacteria, damaged host cells also release N-formylated peptides as damage-associated molecular patterns (DAMPs) [61].

These peptides thus represent a host danger signal, which induces innate immunity activation via specific pattern recognition receptors, corresponding to formyl peptide receptors—named according to their high affinity for N-formylated peptides.

### 2.4. More Than N-Formylated Peptides: Other Ligands Detected by FPRL1

FPRs were originally discovered as receptors with high binding affinity for bacteria or mitochondria N-formylated peptides. In addition to N-formyl-methionine-leucyl-phenylalanine (fMLF), the prototype for N-formylated peptides [18,21], a variety of structurally different ligands have been shown to interact with formyl peptide receptors.

Compared to FPR1, FPRL1 exhibits a lower affinity for formylated peptides. On the contrary, FPRL1 shows the greatest ligand promiscuity, also detecting non-formylated peptides, synthetic molecules, and the eicosanoid Lipoxin A4 (LXA4) [20]. WKYMV is a synthetic, non-formylated peptide recognized by FPRL1. Figure 1 shows the interaction between FPRL1 and WKYMV, displaying the FPRL1 binding site. According to the nature of its ligand, FPRL1 modifies its conformation and consequently its biological function, eliciting a pro- or anti-inflammatory response. In addition, anti-inflammatory ligands may cause receptor homodimerization or heterodimerization with other FPRs, resulting in the activation of the pro-resolving pathway p38/MAPKAPK/Hsp27/IL10, and in the resolution of the inflammation [20,62,63]. The ability of FPRL1 to bind ligands, mediating opposite biological effects, makes it a potential target to control inflammation and inflammatory-associated diseases.

## 3. Formyl Peptide Receptors in *Helicobacter pylori* Chronic Infection

### 3.1. Helicobacter pylori and Chronic Inflammatory Response

*Helicobacter pylori* is a Gram-negative bacterium colonizing the gastric mucosa of over 50% of the population worldwide [64]. The bacterium colonizes the stomach and infects gastric epithelial cells, promoting chronic inflammation, leading to chronic gastritis, which can eventually degenerate in peptic ulcer and then gastric carcinoma [65]. Several studies have reported the relationship between *H. pylori* infection and gastric cancer, classifying the bacterium as the primary cause of gastric carcinoma [66]. Our recent work demonstrates the capability of *H. pylori* to interfere also with biological processes outside the stomach, contributing to the development of neurodegenerative diseases, such as Alzheimer, or metabolic pathologies (type 2 diabetes, obesity, and cardiovascular diseases) [67]. Although the mechanisms responsible for the extra-gastric manifestations of *H. pylori* remain unclear, a plausible explanation may reside in the *H. pylori*-associated, low-grade inflammatory state. Therefore, the clinical outcome of *H. pylori* infection is related to the severity of the inflammatory response, influenced by both host characteristics and bacterial virulence factors [65].

*H. pylori* utilizes various virulence factors responsible for its pathogenicity by inducing different pathways, resulting in cytokine and chemokine release, as well as the production of oxygen/nitrogen species (ROS/RNS) and growth factors [68]. These factors—especially in chronic conditions—contribute to tissue damage and severe disease progression.

Lipopolysaccharide (LPS), peptidoglycan, and the cytotoxic-associated gene A (CagA) are the most studied *H. pylori* pathogenicity factors [68,69]. The gene *CagA* encodes the cytotoxic protein CagA, whose intracellular signaling results in the activation of inflammatory genes and in the modification of the cell scaffold, thus promoting neoplastic transformation [70]. LPS and peptidoglycan instead are directly involved in the bacterium’s adhesion to the gastric epithelium [64]. As these structures are unique to the pathogen, known as pathogen-associated molecular patterns (PAMPs), they are recognized by pattern recognition receptors (PRRs) displayed by eukaryotic cells. Specifically, Toll-like receptor 4 (TLR4) binds LPS, while peptidoglycan is recognized by gastric cells through nucleotide oligomerization domain 1 (NOD1) [68].

FPRs are one of the most relevant class of PRRs. They play an important role in *H. pylori* infection by interacting with Hp(2-20), an *H. pylori*-released, non-formylated peptide with the greatest affinity for formyl peptide receptor like-1. Hp(2-20) is a chemotactic factor [34], contributing to *Helicobacter pylori*-induced gastric inflammation [71,72].

### 3.2. Helicobacter pylori Hp(2-20) Modulated the Host Immune Response by Interacting with FPRL1

Hp(2-20) is a cecropin-like peptide released during *H. pylori* growth and possessing different functional characteristics. As an antimicrobial peptide (AMP), Hp(2-20) plays an important role in *H. pylori* gastric mucosa persistence by acting as bactericidal molecule [73], thus conferring an advantage over other microorganisms. Moreover, it orchestrates the host inflammatory response by interacting with FPRs [34]. When *H. pylori* colonizes the human stomach, the epithelial cells of the gastric mucosa represent the first component of the innate immune response to be encountered [74]. Hp(2-20) participates to activate the innate immune response by interacting with the FPRL1 expressed on epithelial gastric cells. In particular, the interaction between Hp(2-20) and FPRL1 favors *H. pylori* gastric mucosa inflammation [71] by initiating a cell signaling cascade, which leads to cytokine release, the recruitment of inflammatory cells, and stimulation of NADPH oxidase-dependent superoxide generation [75]. This bacterial peptide also stimulates gastric epithelial cell migration and proliferation, and increases the expression level of vascular endothelial growth factor (VEGF), whose role is to restore the gastric mucosa after induced injury [7,34]. These events, if contained, favor the host defense and the restoration of the homeostasis. However, in the case of *H. pylori* infection, the inflammatory response is sustained and exacerbated, thus resulting in the worsening of the clinical outcome due to the onset of severe gastric and extra-gastric diseases [67,76]. Interestingly, despite the massive inflammatory response, *Helicobacter pylori* has developed several strategies to manipulate the host immune system and persist within the gastric mucosa. *H. pylori* is able to both send and integrate signals from the gastric epithelium, allowing the host and bacteria to become linked in a dynamic equilibrium [77]. The long-term persistence of *H. pylori* promotes a permanent and low-grade inflammation, which is a risk factor for severe diseases, including cancer.

Given this scenario, Hp(2-20) has a critical role in the pathological processes associated with *H. pylori* infection by its interaction with FPRL1.

### 3.3. Hp(2-20) and FPRL1: Intracellular Signalling Cascade

In addition to G-protein coupled receptors, after ligand binding, FPRs undergo a conformational change that makes them able to interact with the heterotrimeric G protein. Upon activation, the G protein α subunit exchanges guanosine diphosphate (GDP) with guanosine triphosphate (GTP) and dissociates from β-γ subunits [78], leading to phospholipase Cβ (PLCβ) and phosphoinositide 3-kinaseγ activation (PI3Kγ) [22]. Both phospholipase Cβ and phosphoinositide 3-kinaseγ mediate signaling events associated with different cellular responses, such as chemiotaxis, reactive oxygen species (ROS) generation, and degranulation.

Phospholipase Cβ hydrolyzes phosphatidylinositol 4,5-biphosphate (PIP2) in diacylglycerol (DAG)—which remains at the membrane level—and inositol trisphosphate (IP3). This latter moves to the endoplasmic reticulum, regulating Ca++ release and increasing cytosolic Ca++ levels. In addition, both of the second messengers activate protein kinase C (PKC), resulting in NADPH oxidase activation and ROS production [78].

Phosphoinositide 3-kinaseγ, instead, mediates the conversion of phosphatidylinositol 4,5-biphosphate (PIP2) in phosphatidylinositol 3,4,5-trisphosphate (PIP3), which in turn activates protein kinase C (PKC). Furthermore, PI3Kγ activates protein kinase B (Akt) [79], directly involved in regulating the transcriptional activity of the nuclear factor kB (NF-kB). Other intracellular effectors in the formyl peptide receptors signaling cascade include phospholipase A2 and D, mitogen-activated protein kinase (MAPK), extracellular signal-regulated kinase (ERK)1/2, c-Jun N-terminal kinase (JNK), and p38 [21,80,81].

The Hp(2-20)–FPRL1-induced signaling pathway promotes the activation of p42/44 MAPK (ERK), Akt, and signal transducer and activator of transcription (STAT) 3 [75,82], all mediators with a central role in the host defense against an injury. Akt and ERK induce the activation of the transcriptional nuclear factor kB (NF-kB) [83,84], which plays a critical role in innate immunity as the primary regulator of inflammatory response by activating cytokine genes and the inflammasome [85]. Signal transducer and activator of transcription (STAT) 3 also has important immunomodulatory properties. However, chronic activation of STAT3, due to the *H. pylori* persistence, can degenerate in chronic inflammation and severe associated diseases [86]. Furthermore, several studies have shown the common involvement of the MAPK/ERK pathway in cancer progression [87]. Taken together, these traits make Hp(2-20) and formyl peptide receptor-like 1 critical players in *H. pylori* infection (Figure 2).

Shortly, despite the role of Hp(2-20) in modulating the immune response against *H. pylori* infection, the chronic stimulation of FPRL1 induced by bacterium persistence activates intracellular responses, resulting in exacerbation of the inflammatory process, thus contributing to the clinical outcome of the infection.

## 4. Conclusions and Future Perspectives

The FPR family is one of the most important class of cell surface receptors, playing important roles in innate immunity and host defense by mediating key events during the inflammatory response. Nevertheless, detrimental effects can result from FPR activation. In recent years, several studies have demonstrated the link between FPR activation and the pathogenesis of inflammatory diseases, including cancer [13].

*Helicobacter pylori* infection represents the most frequent cause of gastric carcinoma. It colonizes and infects gastric epithelial cells by promoting chronic inflammation, leading to severe site-specific diseases [88]. In this context, FPRs—specifically FPRL1—play a critical role by interacting with the *H. pylori*-derived peptide Hp(2-20).

The activation of FPRL1 by Hp(2-20) triggers epithelial cell migration, proliferation, and angiogenesis. Thus, chronic stimulation of FPRL1 by Hp(2-20), resulting in persistent low grade inflammation, makes these functions harmful and Hp(2-20) a recognized risk factor, not only for *H. pylori*-associated gastric cancer [89], but also for other chronic inflammatory-related diseases [90].

Based on these considerations, FPR inhibition could represent a promising therapeutical strategy for treating various chronic inflammatory pathologies. Of particular interest is FRPL1, expressed by a variety of cells and tissues and able to bind a broad range of structurally different ligands, exerting pro- or anti-inflammatory actions. Thus, FPRL1 represents a potential target to inhibit inflammatory responses by using agonists with anti-inflammatory properties or antagonists.

This drew attention in searching novel natural molecules able to antagonize FPRL1 or elicit anti-inflammatory responses, paving the way for the discovery of new drugs for the treatment of inflammatory diseases.

Such an approach could also be extended to inflammatory disorders associated with viral infections. Our predictive studies (data not shown) indicate that FPRL1—showing an effective interaction with various coronavirus peptides—could also contribute to the coronavirus disease (COVID)-19 pathogenesis.

## Figures and Tables

**Figure 1 ijms-22-03706-f001:**
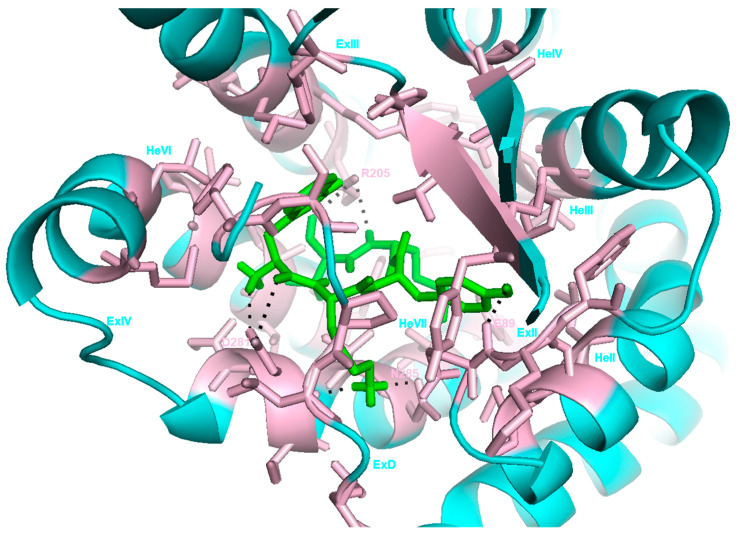
Schematic representation of the synthetic peptide WKYMV binding pocket for FPRL1.The receptor is shown in cartoon representation colored in cyan, while the ligand is shown in sticks colored in green. The amino acids of the binding site are represented in sticks colored in light pink. WKYMV forms hydrophobic interactions with the amino acids of the extracellular domain (ExD); II, III, IV, VI, VII transmembrane portions (He); II, III, IV extracellular portions (Ex); and hydrogen bonds with E89 (ExII), R205 (ExIII), D281, and N285 (ExIV). PyMOL Molecular Graphic System (Version 1.3 Shrodinger, LLC) was used to represent the WKYMV–FPRL1 interaction (PDBcode: 6LW5).

**Figure 2 ijms-22-03706-f002:**
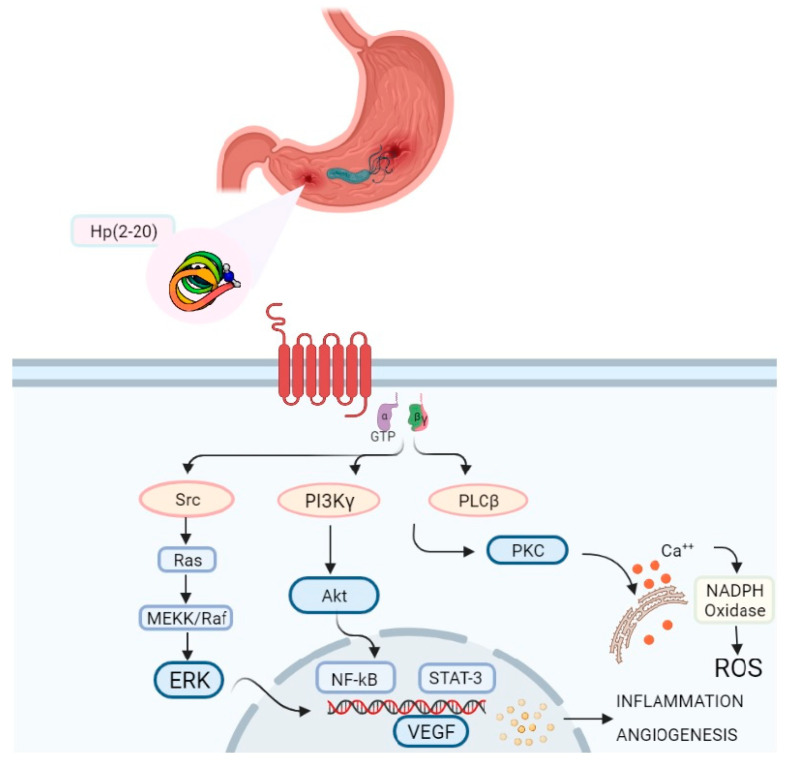
Schematic signalling pathway of Hp(2-20)-activated FPRL1. FPRL1 participates in the pathogenesis of *H. pylori* infection by interacting with Hp(2-20). During Hp(2-20) binding, the G protein α subunit exchanges guanosine diphosphate (GDP) with guanosine triphosphate (GTP) and dissociates from β-γ subunits. This activates various signal transduction events, resulting in oxidative stress, cell migration, inflammation, cell proliferation, and angiogenesis. These events, which are chronically induced, may lead to severe gastric diseases, including cancer. Abbreviations: Akt: protein kinase B; ERK: extracellular signal-regulated kinase; GTP: guanosine triphosphate; MEKK: mitogen-activated protein kinase kinase; NF-kB: nuclear factor kB; PI3Kγ: phosphoinositide 3-kinase gamma; PLCβ: phospholipase Cβ; PKC: protein Kinase C; ROS: reactive oxygen species; STAT3: signal transducer and activator of transcription 3; VEGF: vascular endothelial growth factor.

**Table 1 ijms-22-03706-t001:** Distribution and roles of formyl peptide receptors (FPRs; FPR1, formyl peptide receptor-like (FPRL)1, and FPRL2) in innate, adaptative, and non-immune cells. The table also reports the formyl peptide receptor expression in organs or tissues, or where it is not well-defined.

	Cells/Tissues	Formyl Peptide Receptors	Role	References
Innate Immune Cell Expression	Neutrophils	FPR1, FPRL1	Chemotaxis, phagocytosis, superoxide generation	[16,21,22]
	Natural killer cells	FPR1, FPRL1	Interferonγ production	[16,23]
	Immature dendritic cells	FPR1, FPRL2	Chemotaxis	[15,21,22]
	Mature dendritic cells	FPRL2	Chemotaxis	[15,21,22]
	Monocytes	FPR1, FPRL1, FPRL2	Chemotaxis, pro-inflammatory activity	[16]
	Macrophages	FPR1, FPRL1, FPRL2	Chemotaxis, pro-inflammatory activity	[16]
Adaptative Immune Cell Expression	Naïve CD4 T cells (CD3+, CD4+, CD45RA+, CD45RO–, CCR7+)	FPRL1	Interferon-γ production	[16,17]
	Th1 cells	FPRL1	–	[16,17]
	Th2	FPRL1	–	[16,17]
	Th17	FPRL1	–	[16,17]
Non-Immune Cells, Organ/Tissue Expression	Epithelial cells	FPRL1	Chemotaxis	[18,24]
	Endothelial cells	FPRL1	Chemotaxis, angiogenesis, and cell proliferation	[25]
	Microglial cells	FPRL1, FPRL2	Inflammation and neurogenerative activity	[18,26]
	Keratinocytes	FPRL1	Cell proliferation and pro-inflammatory activity	[27]
	Fibroblasts	FPRL1	Chemotaxis and innate immune response stimulation	[28]
	Astrocytes	FPRL1	Inflammation and neurogenerative activity	[18,26]
	Hepatocytes	FPRL1	Chemotaxis, angiogenesis	[18,29]
	Intestinal epithelial cells	FPR1, FPRL1	Cell proliferation, inflammation, and tumorigenesis	[30]
	Brain	FPRL1	Inflammation and neurodegenerative activity	[16,17,18]
	Spleen	FPRL1, FPRL2	Innate immune response	[16,17,18]
	Placenta	FPRL1, FPRL2	Innate immune response	[16,17,18]
	Lung	FPRL1, FPRL2	Innate immune response	[16,17,18]
	Testis	FPRL1	Innate immune response	[16,17,18]
	Trachea	FPRL2	Innate immune response	[16,17,18]
	Lymph nodes	FPRL2	Innate immune response	[16,17,18]

**Table 2 ijms-22-03706-t002:** Formyl peptide receptor (FPR1, FPRL1, and FPRL2) representative ligands. The table summarizes the main differently derived formylated or non-formylated FPR ligands, indicating the origin, selectivity, and general intracellular signaling.

Classification	Ligand	Origin	Signaling	Selectivity	References
Formylated Bacterial Peptides	f-MLF	*E. coli*	Ca^++^ mobilization, superoxide generation	FPR1	[14]
	f-MKNFKG	*Bacillus*	Ca^++^ mobilization, superoxide generation	FPRL1	[31]
	f-MGFFIS	*Streptococcus*	Ca^++^ mobilization, superoxide generation	FPR1, FPRL1	[31]
	f-MAMKKL	*Salmonella*	Ca^++^ mobilization, superoxide generation	FPR1	[31]
	f-MVMKFK	*Haemophilus*	Ca^++^ mobilization, superoxide generation	FPR1, FPRL1	[31]
	f-MFIYYCK	*Staphylococcus*	Ca^++^ mobilization, superoxide generation	FPR1	[31]
	f-MKKIML	*Listeria*	Ca^++^ mobilization, superoxide generation	FPR1, FPRL1	[31]
	f-MKKNLV	*Clostridium*	Ca^++^ mobilization, superoxide generation	FPRL1	[31]
Formylated Mitochondria Peptides	f-MMYALF	Mitochondrion	Superoxide generation	FPRL1	[32]
	f-MLKIV	Mitochondrion	Ca^++^ mobilization, ERK activation	FPRL1	[32]
	f-MYFINILTL	Mitochondrion	Ca^++^ mobilization, ERK activation	FPRL1	[32]
	f-MFADRW	Mitochondrion	Ca^++^ mobilization, ERKs activation	FPRL1	[32]
	Mitocryptide-2	Mitochondrion	Ca^++^ mobilization, ERK activation	FPRL1	[33]
Microbe-Derived Non-Formylated Peptides	Hp(2-20)	*Helicobacter pylori*	Superoxide generation, cell proliferation, Akt and STAT3 activation, VEGFA secretion	FPRL1	[34]
	OC43 Coronavirus protein	OC43 Coronavirus	Unknown	Not clear	[35]
	229E Coronavirus protein	229E Coronavirus	Unknown	Not clear	[35]
	NL36 Coronavirus protein	NL36 Coronavirus	Unknown	Not clear	[35]
	spike protein	Ebola virus	Unknown	Not clear	[35]
	T20/DP178	HIV gp41	Ca^++^ mobilization	FPR1	[20]
	T21/DP107	HIV gp41	Ca^++^ mobilization	FPR1, FPRL1	[20]
	V3 peptide	HIV gp120	Ca^++^ mobilization, CCR5 desensitization	FPRL1	[36]
	N36 peptide	HIV gp41	Ca^++^ mobilization, chemokine receptorsn desensitization, NF-kB activation	FPRL1	[37]
	gG-2p20	Herpes simplex virus	Superoxide generation, NADPH oxidase activation, apoptosis	FPR1	[38]
	C5a HCV peptide	Hepatitis C virus	Ca^++^ mobilization, degranulation, superoxide generation, MAPK activation	FPRL1	[39]
Host-Derived Molecules	Annexin 1	Host	ERK phosphorylation, NF-kB pathway	FPRL1	[40]
	Lipoxin-A4	Host	Ca^++^ mobilization, ERKs, PI3K, Akt phosphorylation	FPRL1	[41]
	SAA	Host	Ca^++^ mobilization, ERKs, JNK and p38MAPK activation, cytokine release, NF-kB and COX2 induction	FPRL1	[42]
	Aβ-42	host	PI3K/Akt pathway activation	FPRL1	[26]
	LL-37	host	Ca^++^ mobilization, Bcl-xL expression, caspase-3 inhibition, MAPK and JAK/STAT signaling	FPRL1	[43]
Synthetic Peptides	WKYMVm	synthetic	Ca^++^ mobilization, NADPH oxidase activation, ERK phosphorylation, MAPK and JNK activation, PKC activation	FPRL1	[44]
Synthetic Molecules	Quinazolinones	Synthetic	Ca^++^ mobilization, ERK activation	FPRL1	[45]
	Benzimidazoles	Synthetic	Ca^++^ mobilization	FPR1	[45]
	Pyrazolones	Synthetic	Ca^++^ mobilization, desensitization of chemokine receptors	FPRL1	[45]
	Pyridazin-3(2H)-ones	Synthetic	Ca^++^ mobilization	FPR1	[45]
	Chiral pyridazines	Synthetic	Ca^++^ mobilization	FPR1, FPRL1	[45]
	N-phenylureas	Synthetic	Ca^++^ mobilization	FPRL1	[45]

ERK: extracellular signal-regulated kinase; Akt: protein kinase B; STAT3: signal transducer and activator of transcription 3; NADPH: nicotinamide adenine dinucleotide phosphate; VEGF: vascular endothelial growth factor; Aβ-42: β amyloid protein 42; JNK: c-Jun N-terminal kinase; PKC: protein kinase C; COX2: cyclooxygenase-2; PI3K: phosphoinositide 3-kinase; SAA: serum amyloid protein A; CCR5: C–C chemokine receptor 5; JAK: Janus kinase; MAPK: mitogen-activated protein kinase; NF-kB: nuclear factor kB.

## Data Availability

Not applicable.

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
