# Peer review of "The Role of Formyl Peptide Receptors in Permanent and Low-Grade Inflammation: Helicobacter pylori Infection as a Model"

_ijms, 2021, doi:10.3390/ijms22073706_

Round 1

Reviewer 1 Report

In this review, the authors summarized the role of Formyl peptide receptors (FPRs) in Helicobacter pylori associated chronic inflammation. Although the story is appealing, the authors should improve it by adding enough details. The major concerns are listed below.

  1. Please provide a schematic diagram of FPRs protein structure and highlight their binding domains.
  2. Please summarize the roles of FPRs in different cells types, their binding ligands and subsequent signaling under various pathophysiological conditions.
  3. The authors suggest that FPRs could be used as potential target for a new anti-inflam-matory approach; therefore I would like to know if there are any preclinical and clinical trials ongoing.
  4. English editing is required for this manuscript.

Author Response

Comments and Suggestions for Authors

In this review, the authors summarized the role of Formyl peptide receptors (FPRs) in Helicobacter pylori associated chronic inflammation. Although the story is appealing, the authors should improve it by adding enough details. The major concerns are listed below.

Please provide a schematic diagram of FPRs protein structure and highlight their binding domains.

We thank the reviewer for his/her suggestion. As the reviewer suggests, a figure showing the FPR protein- specifically FPRL1- and the binding site referred to a selective agonist has been added (Figure 1). Please see lines 184-186 and 193-201.

Please summarize the roles of FPRs in different cells types, their binding ligands and subsequent signaling under various pathophysiological conditions.

We thank the reviewer for his/her suggestion. As suggested, we have modified the Table 1, by adding a further column in which we have summarized the role of Formyl peptide receptors in different cell types, with corresponding references. Please, see lines. …In addition, a second table (Table 2) has been created, in order to review the different FPRs binding ligands and subsequent signalling. Please, see lines 81; 108-11 and 112-121.   

The authors suggest that FPRs could be used as potential target for a new anti-inflam-matory approach; therefore I would like to know if there are any preclinical and clinical trials ongoing.

We thank the reviewer for raising this point. Biological significance of Formyl peptide receptors and their involvement in several chronic inflammatory and infectious diseases make them clinically relevant. Several studies examined the anti-inflammatory role of formyl peptide receptors novel ligands, by using in vitro model. However, few ligands have been tested by using in vivo animal model and even less have entered clinical phase. Act-389949 is a small molecule recognized by FPRL1, whose potence and stability make it suitable for clinical trials (Biochemical Pharmacology 2019; 166:163).

English editing is required for this manuscript.

We thank the reviewer for his/her comment. English has been improved.

Reviewer 2 Report

The paper is novel and well written.

I have a  request

Can the authors describe the roles of FPRs in the gastrointestinal tract about H. pylori and eventually associated diseases?

Author Response

Comments and Suggestions for Authors

The paper is novel and well written.

I have a  request

Can the authors describe the roles of FPRs in the gastrointestinal tract about H. pylori and eventually associated diseases?

We thank the reviewer for arising this point. Due to its features, the gastrointestinal tract is exposed to numerous pathogens every day. This suggests the importance of the innate immunity, specifically of Pattern Recognition Receptors (PRRs), in ensuring the gastrointestinal homeostasis by mounting a functional immune response. Formyl peptide receptors are PRRs widely expressed at gastrointestinal level, playing a critical role during H. pylori infection. As chronic bacterium, H. pylori promotes a prolonged host defensive response, resulting in chronic inflammation, causative factor for several diseases including cancer. Inflammation and in particular the non-self-limited one, stimulates cell transformations leading to severe damages. Therefore, despite the recognized immunomodulator role of the formyl peptide receptors, they also participate in mediating the mechanisms responsible for cell transformations, promoting gastric cancer or oesophageal adenocarcinoma. Moreover, our previous studies have also brought out the link between H. pylori-chronic inflammation and neurodegenerative or metabolic diseases, suggesting the potential contribute of formyl peptide receptors in cellular processes associated with them.  

Round 2

Reviewer 1 Report

The authors have addressed all my comments.